# A Meta-Analysis of Observational Studies on Prolactin Levels in Women with Polycystic Ovary Syndrome

**DOI:** 10.3390/diagnostics12122924

**Published:** 2022-11-23

**Authors:** Marzieh Saei Ghare Naz, Maryam Mousavi, Fatemeh Mahboobifard, Atrin Niknam, Fahimeh Ramezani Tehrani

**Affiliations:** 1Reproductive Endocrinology Research Center, Research Institute for Endocrine Sciences, Shahid Beheshti University of Medical Sciences, Tehran P.O. Box 19395-476, Iran; 2Department of Biostatistics, Faculty of Medical Sciences, Tarbiat Modares University, Tehran P.O. Box 14115-134, Iran; 3Department of Pharmacology, School of Medicine, Shahid Beheshti University of Medical Sciences, Tehran P.O. Box 1985717443, Iran

**Keywords:** polycystic ovary syndrome, prolactin, meta-analysis, diagnosis, Rotterdam criteria

## Abstract

Women with polycystic ovary syndrome (PCOS) are reported to have different levels of prolactin (PRL) compared to women without PCOS. This study aimed to evaluate the PRL levels in women with PCOS, compared to the control group, before and after adjustment for potential confounders. Using a logical combination of keywords, a comprehensive search was carried out in PubMed and Web of Science, from inception to 30 August 2022. Weighted mean differences (WMDs) with corresponding 95% CIs in PRL levels were employed with a random-effects model. I^2^ was applied to evaluate heterogeneity among studies. A meta-regression analysis and subgroup analysis were conducted to explore heterogeneity sources. Publication bias was assessed by the Egger test. Thirty-two studies, measuring PRL levels in 8551 PCOS patients according to the Rotterdam criteria and 13,737 controls, were included in the meta-analysis. Pooled effect size suggested that the overall weighted mean difference (WMD) of PRL level was significantly higher in women with PCOS, compared to controls (WMD = 1.01, 95% CI: 0.04–1.98, *p* = 0.040). The result of meta-regression adjusted for age, BMI, and the continent of origin, revealed no confounding effect on results. Sub-group analysis of PRL levels according to the continent of origin showed significantly higher PRL levels among Eurasian PCOS patients compared to the control; this difference was not statistically significant in the subgroups of women from Asia, Europe, and South America. In conclusion, PRL levels in patients who were diagnosed according to the Rotterdam criteria were significantly higher than non-PCOS participants. Slightly higher levels of PRL could be presented as a diagnostic feature of PCOS.

## 1. Introduction

Polycystic ovary syndrome (PCOS) is the most frequent endocrine disorder during the premenopausal period; it is considered to be a multicomponent and polygenic disorder [1]. The global prevalence of PCOS constitutes 4 to 20% [2,3]. According to the Rotterdam criteria, PCOS is characterized by any two of the three following features: polycystic ovarian morphology (PCOM), anovulation, and hyperandrogenism, after excluding hyperprolactinemia and any endocrinopathy related to thyroid, pituitary, and adrenal glands, which mimic PCOS [4].

Hyperprolactinemia is the most common endocrine disorder of the hypothalamic-pituitary axis and has a prevalence of approximately 90 per 100,000 women [5]. The Endocrine Society Clinical Practice Guideline has categorized PCOS as an etiology of hyperprolactinemia [5], as 3–67% of women with PCOS suffer from hyperprolactinemia [6]. It is proposed that in women with PCOS, the capacity for estradiol (E2) production was enhanced due to the stimulatory effects of recombinant human follicle-stimulating hormone (r-Hfsh), which may consequently lead to an elevation of PRL [7]. Additionally, inappropriate luteinizing hormone (LH) and PRL secretion in women with PCOS might be related to the low dopamine (DA) hypothalamic tone in women with PCOS [8]. Moreover, PRL levels have been shown to be positively associated with sex hormones such as estradiol and total testosterone [9].

A revised 2003 consensus on diagnostic criteria of PCOS emphasized that the upper normal limit or slightly above normal PRL levels might be observed in the majority of women with hyperandrogenism [10]. The causality of the association between higher levels of PRL and PCOS is still unclear, the higher levels of prolactin could inhibit ovulation and lead to the polycystic ovarian morphology [11]. Despite the existence of evidence in terms of the association between PRL and PCOS [12,13], it is suggested that in cases of co-existence of PCOS and hyperprolactinemia, investigation of the classical etiologies of PRL abnormality is necessary [7]. This lack of further investigation into the etiology of higher PRL levels, as well as guideline modifications for the diagnosis of PCOS through the years, has brought us uncertainty regarding the PRL levels in PCOS.

Studies investigating serum concentrations of PRL in women with PCOS reported inconsistent results; while some of them revealed an elevation of PRL levels in women with PCOS [14,15,16], others reported no significant differences in PRL levels between women with PCOS and healthy women [17,18,19,20]. It is important to note that the majority of available studies had small sample sizes and included women referred to a clinic, hospitalized women, infertile or only included normoprolactinemic women with PCOS [14,15,16,17,18,19,20].

In the present meta-analysis, we aimed to determine whether PRL concentrations in women diagnosed with PCOS according to the Rotterdam criteria are different from their healthy counterparts.

## 2. Materials and Methods

This study was conducted in accordance with the PRISMA statement (Preferred Reporting Items for Systematic Reviews and Meta-Analyses) [21].

### 2.1. Search Strategy

A comprehensive search was conducted up to 30 August 2022, in PubMed and Web of Science. The search terms used were (“Polycystic Ovary Syndrome” OR “Ovary Syndrome, Polycystic” OR “Syndrome, Polycystic Ovary” OR “Stein-Leventhal Syndrome” OR “Stein Leventhal Syndrome” OR “Syndrome, Stein-Leventhal” OR “Polycystic Ovarian Syndrome” OR “Ovarian Syndrome, Polycystic” OR “PCOS”) AND (“Hyperprolactinemia” OR “Prolactin, Inappropriate Secretion” OR “Inappropriate Secretion Prolactin” OR “Secretion Prolactin, Inappropriate” OR “Prolactin Hypersecretion Syndrome” OR “Hypersecretion Syndrome, Prolactin” OR “Syndrome, Prolactin Hypersecretion” OR “Inappropriate Prolactin Secretion” OR “Prolactin Secretion, Inappropriate” OR “Prolactinoma” OR “HPRL” OR “Prolactin” OR “PRL”). We restricted our search to the English language. An additional manual retrieval was conducted through reference lists of the selected studies to detect other relevant studies.

### 2.2. Inclusion Criteria

Studies meeting the following inclusion criteria were included: (1) observational studies, including cross-sectional, case-control, and cohort studies with data on confirmed PCOS patients according to Rotterdam diagnosis criteria and non-PCOS controls. (2) Studies with data on serum PRL levels in both PCOS women and controls after excluding the other causes of hyperprolactinemia (including pituitary tumours, hypothyroidism, pregnancy, thyroid disorder, and Cushing syndrome, etc.).

### 2.3. Data Extraction

Two independent authors (M.S.G and F.R.T) extracted the following information from the eligible studies: first author, the date of publication, country, study design, sample size, diagnostic criteria, detection method, the mean and standard deviation/median and interquartile range/median and range of serum PRL concentrations.

If a study presented serum PRL levels through other means of central tendency (quartile, percentile, and median), the values were converted to mean ± SD using proper formulas [22]. The different reported measurement units of PRL in different studies were all converted to a single unit, which was ng/mL.

### 2.4. Quality Assessment

The quality of included articles was assessed and scored by two investigators independently (F.M. and MSG), by applying the Newcastle–Ottawa scale (NOS), to improve the interpretation of the results and reduce the review bias. Scoring the NOS grade was carried out according to three aspects: selection, comparability, and exposure. According to the quality score assessment, each item on the scale is scored from one point, except for comparability, which can be adapted to the specific topic of interest to score up to two points [23]. NOS score categories of 0–3, 4–6, and 7–9 were considered as low, moderate, and high quality, respectively. Disagreements were resolved by discussion with the third reviewer (FRT).

### 2.5. Statistical Analysis

The combined mean difference and the corresponding 95% confidence interval (95% CI) of PRL levels in the PCOS and non-PCOS participants of all included studies were calculated. The statistical significance of pooled mean difference was estimated with the Z test. The graphical results were displayed by forest plot.

The chi-squared test and I^2^ statistic were calculated to evaluate statistical heterogeneity among studies. A random-effects model was performed when significant heterogeneity (I^2^ > 50% or *p* < 0.05) was detected. Further subgroup analysis was conducted to assess the potential sources of heterogeneity. To examine possible heterogeneities in the meta-analysis, a meta-regression analysis was performed using the following variables: age, continent of origin, and BMI. Publication bias was evaluated using Begg’s funnel plot and Egger’s regression test. STATA software version 14.0 (StataCorp LP, College Station, TX, USA) was used to conduct the meta-analysis, and *p* < 0.05 was considered as statistically significant.

## 3. Result

### 3.1. Characteristics of Included Studies

A total of 32 studies (Figure 1) involving 22,288 participants (8551 PCOS and 13,737 non-PCOS) met the inclusion criteria and were included in the meta-analysis. Characteristics of the included studies are presented in Table 1. Thirteen studies were conducted in Asia, six in Europe, two in Africa, one in South America, and ten in Eurasia. In this meta-analysis, according to the quality score assessment, eighteen of the reports were classified as moderate quality, and the other studies (*n* = 14) were classified as high quality (Appendix A).

### 3.2. Differences in PRL Levels between PCOS Patients and Non-PCOS

The results indicated that there is a significant difference between PRL levels in PCOS patients and controls (WMD = 1.01, 95% CI: 0.04–1.98, *p* = 0.04). (Figure 2).

### 3.3. Sub-Grouped Meta-Analysis

Sub-group analysis was performed among infertile subjects and subjects in different geographical regions. When stratified by infertility status (n = 6), the results indicated that there was no significant difference between PRL levels in infertile patients with PCOS and non-PCOS (WMD = −0.910, 95% CI: −1.94–0.12, *p* = 0.764). When stratified by continent, the results indicated that PCOS patients of Asia and Europe did not have significantly different PRL levels than controls (WMD = −0.04, 95% CI: −0.79–0.89, *p* = 0.911, and WMD = 2.50, 95% CI: −0.64–5.65, P = 0.119, respectively). African PCOS patients had significantly lower PRL levels (WMD = −2.95, 95% CI: −4.31–−1.58, *p* < 0.001). Eurasian PCOS patients had significantly higher PRL levels (WMD = 1.79, 95% CI: 0.61–2.98, *p* = 0.003). However, South American PCOS patients had comparable PRL levels with non-PCOS participants (WMD = −1.35, 95% CI: −3.46–0.75, *p* = 0.209) (Figure 3).

### 3.4. Heterogeneity

In the present study, the random-effects model was performed for the following analyses due to the statistically significant heterogeneity (I^2^ = 99.4%, *p*-value < 0.001) among studies (Figure 2). Publication bias was evaluated via conducting the Egger test (Coef: 1.36, CI: 0.77–1.96). Figure 2 presents the forest plot of stratified models by continent in the subgroup meta-analysis.

### 3.5. Meta-Regression Results

Meta-regression analysis results showed that age, BMI, and the continent of origin did not have a significant effect on PRL levels in PCOS patients and controls. The results of meta-regression analysis concerning the effect of confounders on the association between PCOS status on prolactin level are presented in Table 2.

## 4. Discussion

The results of this meta-analysis, which included 32 observational studies conducted in different geographical regions, provided evidence to suggest that PRL levels are significantly higher in women with PCOS compared to non-PCOS women. Furthermore, Eurasian patients with PCOS showed significantly higher PRL levels compared to the control group.

Prolactin (PRL) is a versatile polypeptide hormone that is involved in many physiologic functions, including reproduction, growth and development, metabolism, immunoregulation, brain function, and behavioral regulation [47]. PRL is secreted by pituitary lactotroph cells. The production of PRL is stimulated by thyrotropin-releasing hormone (TRH), estrogen, and dopamine receptor antagonists [5]. Moreover, a variety of pathological, physiological, and genetic conditions can affect lactotroph cells to increase their PRL secretion, leading to hyperprolactinemia. Some of the physiological conditions are pregnancy, lactation, stress, and excessive exercise [48,49]. Pathological conditions include drug-induced hyperprolactinemia (DIH), non-functioning pituitary adenomas (NFPA), primary hypothyroidism, polycystic ovary syndrome (PCOS), chronic renal failure, and liver cirrhosis [5,50]. Another cause of increased PRL levels is macroprolactinemia. Macroprolactin is a large molecule of PRL that mostly binds to antibodies and does not exert the biological effect of PRL [51]. The presence of this molecule can cause decreased renal clearance of prolactin and concomitant increases in serum prolactin levels [52]. This wide spectrum of etiologies presents challenges regarding the diagnosis of hyperprolactinemia; moreover, it might explain why the associations of PRL with some of these conditions are still under investigation, one of them being the link between its levels and PCOS.

The PRL level of women with PCOS is an issue that has been studied for years; however, no definite conclusion has been drawn. Although some studies indicated a mild elevation of PRL levels in women with PCOS [14,53,54], other studies did not support this hypothesis [17,18,19,20]. Some even proposed that the elevation of PRL levels in women with PCOS is a transient phenomenon and is likely related to underlying stress, use of offending drugs, or hypothyroidism [55]. Taken together, such disparities between studies may be a result of their clinical heterogeneity, target population, small sample sizes, and lack of adequate statistical power. Another important reason for this controversy might be the challenges regarding the proper diagnosis of PCOS and hyperprolactinemia.

The present study showed that PRL levels were significantly higher in the PCOS group, while this was not observed in the subgroup of studies with infertile subjects (n = 6). The exact mechanism by which PRL might be associated with PCOS remains unknown; however, several plausible mechanisms have been suggested. There is evidence demonstrating that physical, psychological, and environmental factors could play a role in the regulation of the PRL secretion [56]. Women diagnosed with PCOS are reported to have a higher gonadotropin-releasing hormone (GnRH)/LH pulse frequency, which might be responsible for an increase in PRL levels in PCOS [6]. Several neurotransmitters and neuropeptide receptors have been found in GnRH neurons that are directly involved in the release of GnRH, LH, and FSH [57]. One of these neurotransmitters is dopamine, which is an inhibitor of LH release [58]. As a result, a feature of PCOS is reduced dopaminergic tone, which is linked to increased LH release [59,60]. Dopamine is also an inhibitor of PRL release. Therefore, a positive association between PCOS, low dopamine, and hyperprolactinemia has been suggested [8]. Recently a review study depicted metabolic and stress-related roles of the PRL [61]. While the majority of women with PCOS have a higher level of perceived stress [62], they also suffer from metabolic abnormalities [63]. Even though PRL secretion could be affected by the secretion of cortisol, it could surge following psychological stress in the absence of cortisol surge [64]. A recent meta-analysis showed that cortisol levels were significantly higher in PCOS patients compared to healthy controls [65].

Furthermore, in the present study, PCOS patients in the Eurasian population, but not Asian, European, and South America population, had significantly higher PRL levels. Moreover, PCOS patients in the African population have significantly lower PRL levels. This geographical/racial heterogeneity might have several explanations. It is proposed that photoperiod (day length) influences the production of PRL via endogenous timers that reside inside the pituitary gland, through the secretion of melatonin from the pineal gland. Melatonin might impose direct effects on the hypothalamic pathways that control prolactin secretion [66]. There is also evidence for increased pituitary sensitivity to dopamine under short day length [67], which results in decreased PRL secretion even when dopamine concentrations are low, as suspected in PCOS. Moreover, the development of PCOS is influenced by pre-existing insulin resistance and obesity. A study demonstrated that among pregnant women, higher levels of PRL were associated with reduced glucose tolerance [68]. By contrast, a meta-analysis showed an inverse association between higher levels of PRL within the physiological range and prevalence of T2DM [69]. A recent study among women with PCOS did not reveal any differences between fasting plasma glucose and PRL in the PCOS and the non-PCOS group; while in the PCOS group, PRL positively was associated with fasting plasma glucose [13]. The circulating concentration of PRL acts as a determinant factor on the impact of prolactin on glucose metabolism and insulin resistance [70]. In this meta-analysis, the variations between different geographical regions in terms of PRL levels might lead to variations regarding glucose metabolism. It is assumed that in PCOS, even serum PRL levels within the physiologic range are associated with changes in glucose metabolism [35]. This could lead to different PCOS definitions and different patterns of phenotypes, which are possibly related to different reported levels of PRL secretion in different regions [71,72]. Moreover, as many as 75% of individuals with PCOS remain undiagnosed, resulting in various rates of PCOS diagnosis. Variability in clinical presentation and inadequate provider knowledge may be the main contributing factors [73].

Meta-regression analysis showed that age, BMI, and the continent of origin did not have a significant impact on the heterogeneity between the studies. Age and body fat composition have been considered determinant factors of PRL secretion. PRL release was significantly associated with the size of the visceral fat mass [74]. In other words, PRL secretion is enhanced in obese women due to their higher visceral fat mass. Moreover, a study showed that PRL levels among women slightly decrease from age 20 years to middle age; however, it was slightly higher thereafter [75]. Another study also demonstrated that postmenopausal women exhibited a 40% decrease in PRL, which is related to dropping estrogen levels among postmenopausal women [76]. All included studies were published after consensual diagnosis criteria of PCOS, hence it is proposed that constant improvement in diagnostic criteria and evaluation of PRL was achieved in these studies. However, phenotypic heterogeneity of PCOS might also affect PRL levels among PCOS patients; however, in this study, due to the lack of data on different phenotypes of PCOS, we were unable to analyze it. This study included researches that diagnosed PCOS according to the Rotterdam criteria, which are less restrictive than the NIH criteria, include women with milder forms of PCOS alongside those with severe forms.

Our literature review showed that studies that used other international diagnostic criteria, such as National Health Institute (NIH), and Androgen Excess Society (AES) criteria, showed no significant differences in PRL levels between the PCOS group and controls [77,78,79]. There is variability between clinical presentations of PCOS according to the different diagnostic criteria. In the studies conducted based on Rotterdam criteria, patients presenting with anovulation and polycystic ovaries are also considered to have PCOS, which is not the case in other diagnostic criteria. Moreover, both hyperprolactinemia and PCOS are associated with hyperandrogenism and anovulation, although through separate mechanisms [80]. Therefore, the lower diagnosis rate among patients with anovulation in NIH and AES compared to Rotterdam criteria might be the reason why the studies conducted based on these criteria could not reach statistical significance regarding PRL levels. Furthermore, it is proposed that the Rotterdam criteria can recognize all possible phenotypic combinations, so it might contribute to differences of findings.

### Strengths and Limitations

This study is the first meta-analysis to summarize the evidence regarding the PRL levels in women with PCOS in a systematic manner. Another strength of this study is that meta-regression was implemented to examine possible heterogeneities within the data. Furthermore, we recognize the strength of using a WMD as an effect estimate, because it enhances the clinical interpretation of the results.

This meta-analysis is also subject to limitations. The main limitation is the strict inclusion criteria that restricted us to only including papers comparing PRL levels in women with PCOS and those without; in the absence of a comparison group, no assessment, comparison, and pooling of levels can be performed across different studies. Another limitation is including papers that used a single diagnostic criterion for PCOS (Rotterdam criteria); it is suggested that future reviews include studies implementing different international diagnostic criteria for PCOS. A third limitation is the different methods employed by various studies for the assessment of serum concentration of PRL; and the absence of a comprehensive explanation for the exclusion methods of hyperprolactinemia pathologic causes in the majority of included studies. Moreover, it should be noted that there was significant heterogeneity across included studies regarding the clinical features of PCOS and demographic characteristics of the participants. As discussed earlier, this heterogeneity could result in different patterns of metabolic features associated with PCOS that might affect PRL levels, including differences in the secretion of dopamine and melatonin [59,60,66], as well as differences in the prevalence of obesity and insulin resistance [71,72].

## 5. Conclusions

PRL levels were significantly higher among patients diagnosed with PCOS according to Rotterdam criteria than among non-PCOS participants. It can be argued that slightly higher levels of PRL can serve as a diagnostic component of PCOS, which shows the clinical significance of the present study’s findings. Further studies are necessary to elucidate the mechanism underlying this association.

## Figures and Tables

**Figure 1 diagnostics-12-02924-f001:**
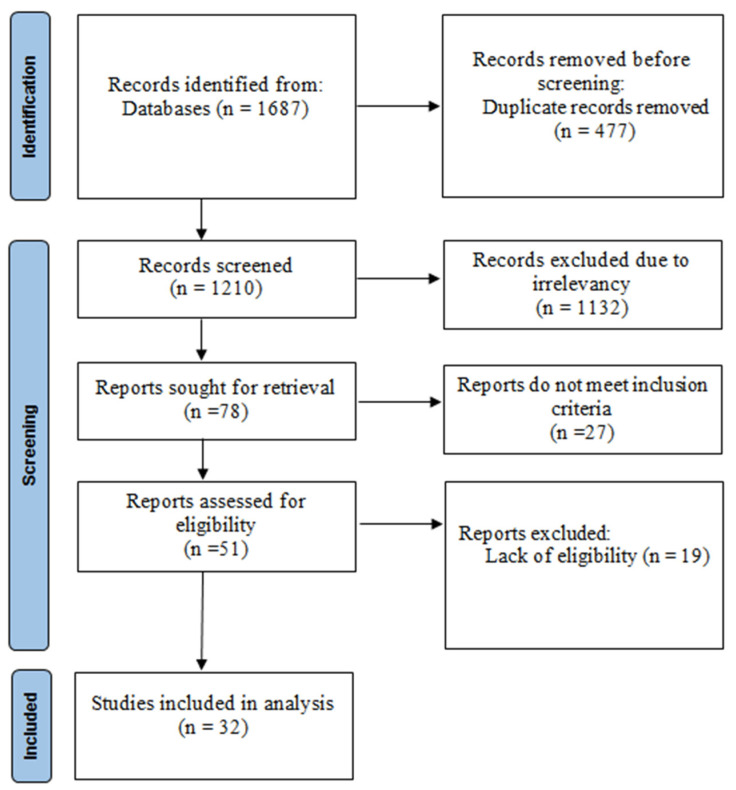
Flowchart of included studies.

**Figure 2 diagnostics-12-02924-f002:**
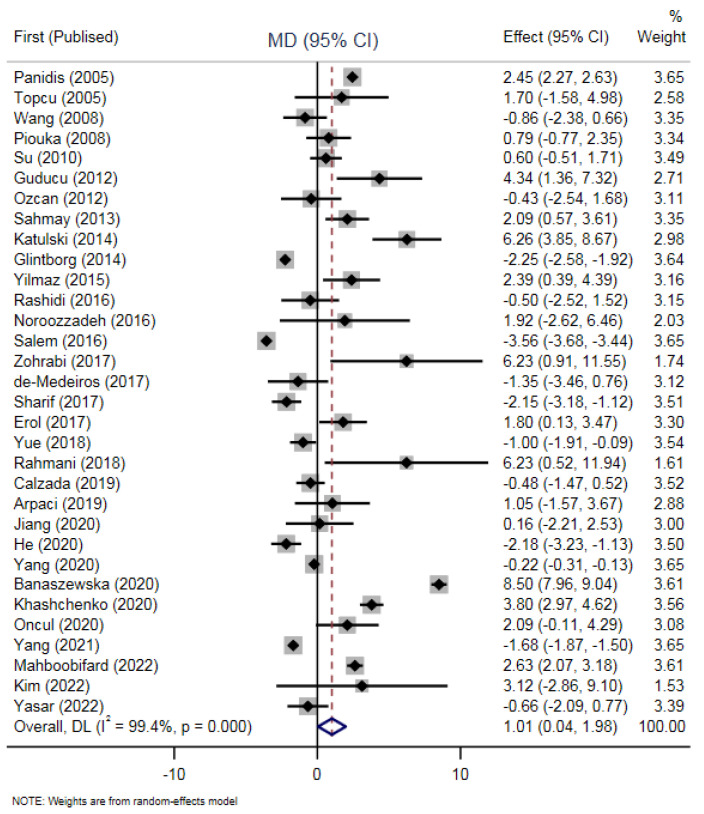
Meta-analysis of 32 comparisons reporting on PRL in PCOS participants compared with non-PCOS [9,13,14,15,16,17,18,19,20,24,25,26,27,28,29,30,31,32,33,34,35,36,37,38,39,40,41,42,43,44,45,46].

**Figure 3 diagnostics-12-02924-f003:**
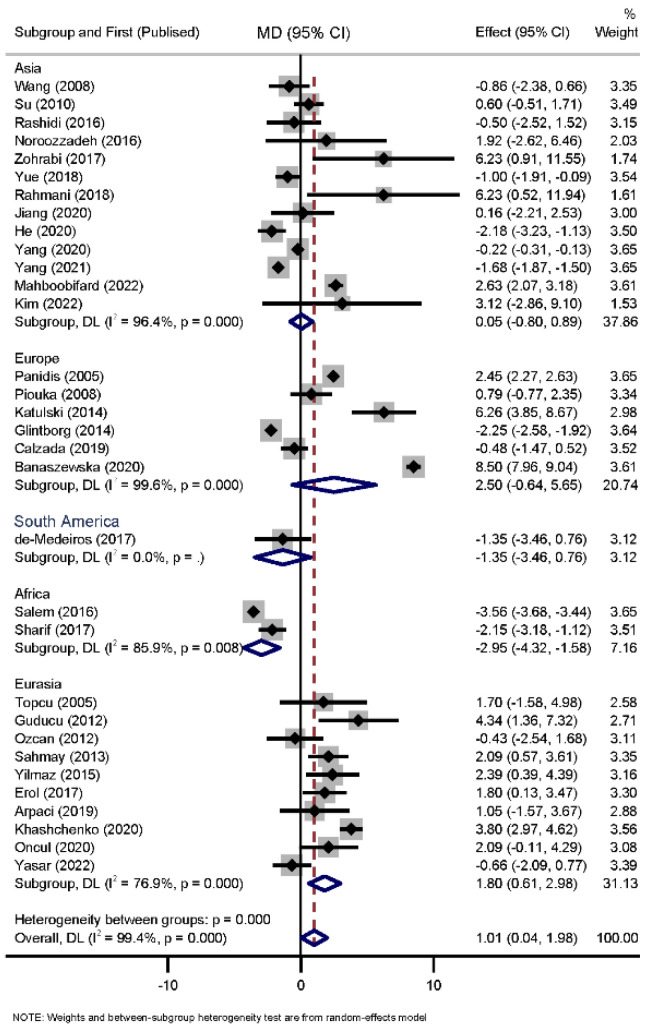
Sub-grouped meta-analysis of 32 comparisons reporting on PRL in PCOS participants compared with non-PCOSs by continent [9,13,14,15,16,17,18,19,20,24,25,26,27,28,29,30,31,32,33,34,35,36,37,38,39,40,41,42,43,44,45,46].

**Table 1 diagnostics-12-02924-t001:** Characteristics of included studies.

First Author, Year (Ref.)	Type of Study	Location	Sample Size		Participants Characteristics PCOS/Non-PCOS	PRL Levels		Assay Methods	Quality
			PCOS	Control		PCOS	Control		
Mahboobifard, 2022 [14]	cross-sectional	Iran	216	702	Age: 31.2 ± 7.9/34.9 ± 7.6 BMI: 26.7 ± 5.4/26.8 ± 5.1	16.1 (10.5–23.6) (ng/mL)	13.4 (9.3–19.7) (ng/mL)	IRMA ^1^	High
Calzada, 2019 [17]	case-control	Spain	77	106	Age: 27 (24–32)/29 (22–33) BMI: 25.3 (20.6–30.7)/21.6 (19.8–23.0)	396.6 (290.5–515.0) (mIU/mL)	371.0 (307.6–589.7) (mIU/mL)	chemiluminescent enzymatically two-site immunoassays	Moderate
Kim, 2021 [18]	cohort	Korea	43	28	Age: 24.91 ± 6.80/27.89 ± 9.57 BMI: 23.19 ± 4.67/21.94 ± 4.57	18.84 ± 15.86 (ng/mL)	15.72 ± 9.84 (ng/mL)	Not clear	High
Salem, 2015 [24]	cross-sectional	Tunisia	118	150	Age: 29.8 ± 0.4/33.5 ± 0.5 BMI: 28.4 ± 0.7/23.1 ± 0.2	73.1 ± 11.7 (mU/L)	148.8 ± 9.4 (mU/L)	Not clear	Moderate
Katulski, 2016 [25]	case-control	Poland	69	30	Age: 23.13 ± 4.43/24.53 ± 2.67 BMI: 23.07 ± 5.9/21.93 ± 1.23	14.98 ± 9.09 (ng/mL)	8.72 ±3.07 (ng/mL)	electrochemiluminescence immunoassay	Moderate
Yasar, 2022 [26]	cross-sectional	Turkey	180	100	Age: 25.94 ± 6.18/28.12 ± 7.27 BMI: 29.77 ± 6.65/28.04 ± 6.16	11.44 ± 5.02 (ng/mL)	12.10 ± 6.25 (ng/mL)	chemiluminescent method	High
Sahmay, 2013 [15]	cross-sectional	Turkey	419	151	Age: 25.82 ± 5.3/26.62 ±5 BMI: 25.43 ± 4.6/25.4 ± 4.4	18.85 ±8.79 (ng/mL)	16.76 ± 7.96 (ng/mL)	Not clear	Moderate
Glintborg, 2014 [9]	cross-sectional	Denmark	1007	116	Age:30 (23–36)/28 (24–37) BMI:27.4 (23.2–33.0)/25 (22.3–29.2)	7 (5–10) (μg/L)	9 (7–13) (μg/L)	two-site commercial kit	Moderate
Jiang, 2020 [19]	cross-sectional	China	93	77	Age: 28.60 ± 3.78/26.62 ± 5 BMI: 24.95 ± 4.05/22.30±3.69	16.82 ± 8.88 (ng/mL)	16.66 ± 6.85 (ng/mL)	electrochemiluminescence	Moderate
Erol, 2017 [20]	case-control	Turkey	60	50	Age: 24.7 ± 3.9/25.8 ± 4.4 BMI: 21.9 ± 1.7/21.8 ± 2.1	12.3 ± 5.1 (ng/mL)	10.5 ± 3.8 (ng/mL)	immunoenzymatic methods	Moderate
Khashchenko, 2020 [16]	case-control	Russia	130	30	Age: 24.7 ± 3.9/25.8 ± 4.4 BMI: 21.9 ± 1.7/21.8 ± 2.1	266.0 (175.0–405.0) (mIU/L)	189.0 (142.0–269.0) (mIU/L)	electro- and immunochemiluminiscent methods	Moderate
He, 2020 [27]	case-control	China	175	196	Age: 30.69 ± 1.7/ 31.00± 1.74 BMI: 21.63 ± 1.23/21.14± 1.14	16.79 ± 4.12 ng/mL	18.79 ± 2.92 ng/mL	radioimmunoassay	High
Yue, 2018 [28]	case-control	China	653	118	Age: 26.9 ± 4.2/27.3 ± 4.1 BMI: 26.2 ± 5.2/25.5 ± 5.6	11 ± 4.9 (ng/mL)	12 ± 4.6 (ng/mL)	Beckman Coulter DxI800	Moderate
Oncul, 2020 [29]	case-control	Turkey	46	46	Age: 24.2 ± 3.6/25.9 ± 4.9 BMI: 27.4 (15.7–38.9)/24.8 (17.6–33.1)	16.87 ± 6.26 ng/mL	14.78 ± 4.31 ng/mL	Electrochemiluminescence method	High
Yang, 2020 [13]	cross-sectional	China	2052	9696	Age: 29.12 ± 0.63/30.95 ± 0.66 BMI: 22.8 ± 1.2/21.18 ± 1.03	11.71 ± 1.92 (mIU/L)	11.93 ± 2.01 (mIU/L)	chemiluminescence assay	Moderate
SU, 2010 [30]	case-control	Taiwan	266	107	Age: 26.2 + 5.4/29.5 + 6.2 BMI: 25.5 + 6.3/22.1 + 5.1	13.5 ± 4.8 (ng/mL)	12.9 ± 5.0 (ng/mL)	EIA ^2^	High
Sharif, 2017 [31]	case-control	Sudan	50	50	Age: 26.9 (5.2)/27.1 (4.8) BMI: 28.4 (4.2)/25.6 (5.7)	12.0 (9.9–17.2) mIU/L	13.9 (10.7–21.2) mIU/L	immunoassay	High
Panidis, 2005 [32]	case-control	Greece	291	109	Age:23.45 ± 0.54/26.74 ± 0.96 BMI:25.51 ± 0.24/25.84 ± 0.505	15.34 ± 0.84 (ng/mL)	12.89 ± 0.83 (ng/mL)	RIA	Moderate
Rashidi, 2016 [33]	cohort	Iran	595	157	Age: 26.94 (4.57)/29.96 (5.91) BMI: 22.27 (4.91)/25.83 (4.45)	14.75 ± 11.79 (ng/mL)	15.25 ±11.4 (ng/mL)	radioimmunoassay	High
Noroozzadeh, 2016 [34]	cross-sectional	Iran	63	216	Age: 33.6 ± 7.2/36.3 ± 6.9 BMI: 27.14 ± 5.74/27.35 ± 4.95	16.98 ± 17.66 (pg/mL)	15.06 ± 9.57 (pg/mL)	IRMA ^3^	High
Yang, 2021 [35]	cross-sectional	China	792	700	Age: 29 (27–32.5)/31 (28–35) BMI:23.73 (21.48-26.85)/21.64 (19.53–23.88)	235.74 (186.85–318.03) (mIU/L)	275.13 (213.60–355.84) (mIU/L)	chemiluminescence	High
Piouka, 2008 [36]	case-control	Greece	200	100	Age:25.18 ± 5.42/26.8 ± 4.65 BMI:26.475 ± 2.8/26.4 ± 3.05	14.54 ± 6.72 ng/mL	13.75 ± 6.4 ng/mL	RIA	Moderate
Yilmaz, 2015 [37]	case-control	Turkey	84	56	Age:22.55 ± 3.45/ 23.5 ± 4.4	15.79 ± 7.34	13.40 ± 4.85	Not clear	Moderate
Güdücü, 2012 [38]	cross-sectional	Turkey	62	40	Age: 24.77 ± 4.85/28.13 ± 5.66 BMI: 24.15 ± 5.35/23.35 ± 5.33	20.87 ± 9.21 (ng/mL)	16.53 ± 6.16 (ng/mL)	Not clear	High
Arpaci, 2019 [39]	case-control	Turkey	46	42	Age: 24.89 ± 6.11/29.02 ± 6.85 BMI: 25.34 ± 5.54/25.00 ± 4.79	13.20 ± 5.52 (ng/mL)	12.15 ± 6.87 (ng/mL)	CMIA ^4^	Moderate
Banaszewska, 2020 [40]	case-control	Poland	62	42	Age: 27.5 ± 0.6/29.1 ± 0.7 BMI: 25.4 ± 0.8/23.3 ± 0.6	22.5 ± 1.6 (ng/mL)	14.0 ± 1.2 (ng/mL)	electrochemiluminescence assays	High
Zohrabi, 2017 [41]	case-control	Iran	30	30	Age: 25.85 ± 5.90/28.91 ± 8.1 BMI: 24.91 ± 3.63/24.02 ± 5.60	21.23 ± 11.94 (ng/mL)	15.00 ± 8.84 (ng/mL)	enzymatic techniques	Moderate
deMedeiros, 2017 [42]	case-control	Brazil	462	232	Age: 26.72 ± 5.38/30.34 ± 4.74 BMI: 29.11 ± 6.74/24.47 ± 4.03	558.02 ± 293.24 (nmol/L)	627.69 ± 287.7 (nmol/L)	electrochemiluminescence assay no bach	Moderate
Wang, 2008 [43]	case-control	China	271	296	Age: 28.84 ± 3.40/29.28 + 3.70 BMI: 24.98 + 4.06/22.19 + 3.03	17.70 ± 9.13 (μg/L)	18.56 ± 9.38 (μg/L)	chemiluminescence immunization	Moderate
Özcan, 2012 [44]	case-control	Turkey	40	35	Age: 22.80 ± 2.70/23.82 ± 1.79 BMI: 27.94 ± 6.78/21.59 ± 2.68	15.5 2± 5.07 (ng/mL)	15.95 ± 4.25 (ng/mL)	electrochemilluminescence immunoassay	Moderate
Rahmani, 2018 [45]	case-control	Iran	26	26	Age: 25.85 ± 5.90/28.91 ± 8.1 BMI: 24.91± 3.63/24.02 ±5.60	21.23 ± 11.94 (ng/mL)	15.00 ± 8.84 (ng/mL)	ELISA ^5^	High
Topcu, 2005 [46]	case-control	Turkey	28	26	Age: 27.1 ± 4.5/28.8 ± 4.4 BMI: 26.6 ± 5.7/24.7 ± 3.7	15.5 ± 7.7	13.8 ± 4.2	chemiluminescent enzymeimmunoassay	High

^1^ Immunoradiometric assay; ^2^ Enzyme immunoassay; ^3^ Immunoradimetric assay; ^4^ Chemiluminescent microparticle immunoassay; ^5^ Enzyme-linked immunosorbent assay.

**Table 2 diagnostics-12-02924-t002:** The results of meta-regression analysis concerning the effect of confounders on the association between PCOS status on prolactin level.

Outcome	Group	Variable *	Regression Coefficient (95% CI)	*p*-Value
PRL	PCOS	Age	−0.21 (−0.91, 0.48)	0.536
		BMI	−0.61 (−1.71, 0.48)	0.264
		Europe	1.49 (−4.54, 7.52)	0.609
		Africa	−6.51 (−13.08, 0.05)	0.052
		South America	13.40 (−16.71, 0.43)	0.351
		Eurasia	1.31 (−2.90, 5.53)	0.524
	Non-PCOS	Age	0.07 (−0.35, 0.49)	0.732
		BMI	0.05 (−0.82, 0.93)	0.901
		Europe	−1.36 (−4.44, 1.70)	0.360
		Africa	−5.07 (−10.21, 0.06)	0.053
		America	13.71 (−10.75, 38.19)	0.246
		Eurasia	−2.16 (−5.60, 1.26)	0.204

Abbreviations: PCOS, Polycystic ovary syndrome; PRL, Prolactin. * Asia is considered as a reference.

## Data Availability

Not applicable.

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
