# Peer review of "A Meta-Analysis of Observational Studies on Prolactin Levels in Women with Polycystic Ovary Syndrome"

_diagnostics, 2022, doi:10.3390/diagnostics12122924_

Round 1

Reviewer 1 Report

The article is very interesting, innovative. 

Author Response

  • Author response: Thank you for your positive feedback and acknowledgment of our manuscripts.

Reviewer 2 Report

Manuscript ID: diagnostics-2011334

Title: Diagnostic Features of Polycystic Ovary Syndrome (PCOS): A Meta-analysis of Observational Studies on Prolactin Levels in Women with PCOS

Authors: Marzieh Saei Ghare Naz, Maryam Mousavi, Fatemeh Mahboubofard, Atrin Niknam, and Fahimeh Ramezani Tehrani 

In this meta-analysis, the authors statistically analyzed data from 32 observational studies comprising 22,288 participants (8,551 PCOS and 13,737 non-PCOS) from Asia, Europe, and Brazil. Using various statistical analysis tools and data mining, the authors reported significantly higher prolactin (PRL) levels in Euro-Asian women with PCOS compared to their non-PCOS controls. Although the authors were successful in their data mining, however, numerous articles related to the preparation of the manuscript, over-generalization of the results together with numerous inaccuracies, inconsistencies, and ambiguities in their reporting precluded this reviewer from accepting this manuscript in its current format.

Major concerns:

-       Title: considering that this meta-analysis was conducted on data exclusively obtained from studies on Asian and European women with PCOS (excluding a solitary South American report from Brazilian), the title should be modified to reflect on this fact (e.g., “A Meta-analysis of Observational Studies on Prolactin Levels in infertile Euro-Asian Women with PCOS”). This meta-analysis report is not about diagnostic features of PCOS! Please rephrase accordingly.

-       Abstract:

o   Line 28: the word “America” should be replaced by “South America” for accuracy and consistency.

-       Introduction:

o   Lines 54- 58: this is an inaccurate statement on the requirement for excluding other aetiological factors for hyperprolactinemia in women suspected of PCOS. Please rephrase accordingly and provide accurate and adequate citations (current reference #7 (Delcour et al (2019)) cannot be considered in drawing such a major conclusion).

o   Lines 66-68: “women” should be replaced with “Euro-Asian women” for consistency and accuracy.

-        Results:

o   Line 130: “America” should be replaced with “South America” for accuracy.

o   Lines 138-139: “PCOS patients” should be replaced with “Euro-Asian PCOS patients” for consistency and accuracy.

o   Figures 2 and 3:

§  It is hard to read the figures. Please replace them with higher-quality images.

§  The X-axis needs to be labeled in both figures. It's unclear what is being measured and/or depicted in these forest plots. Please clearly label the X-axis in these plots.

o   Lines 147- 154: this is very confusing. It's unclear what exact parameters/covariates are being considered in comparing the Asian and European PCOS patients described in lines 151- 154 and those mentioned in lines 147- 149. Also, it’s unclear what population of African women with PCOS compared their PRL levels with (line 150). Please provide further clarifications and rephrase accordingly.

-       Discussion:

o   Lines 170- 174: please remove “across the globe”. Your current meta-analysis does not support the globalization of results.

o   Lines 172- 173: should read: ‘infertile Euro-Asian patients with PCOS”.

o   Lines 175- 192: acknowledging that PCOS is known to alter PRL levels in women, please further discuss the significance of your findings, as well as agreements and disagreements with previous reports accordingly.

o   Lines 202- 203: this is very confusing and contradicts the authors’ conclusions on lines 144- 146 regarding the lack of statistical significance in PRL levels between fertile PCOS and their non-PCOS controls! Please consider rephrasing accordingly.

o    Lines 230- 236: the authors should further elaborate on the influence of insulin resistance on PRL levels in PCOS women in view of their own meta-analysis (e.g., how insulin resistance might have potentially influenced the outcome of your meta-analysis).

o   Lines 253- 256: the authors should further discuss and elaborate on plausible factors contributing to the lack of statistical significance in PRL levels among PCOS women reported through the NIH- and AES- based inclusion criteria (e.g., how NIH and AES inclusion criteria influenced outcomes of statistical analyses of PRL levels in infertile PCOS women?).  

Author Response

# Reviewer 2

Major concerns:

Title: considering that this meta-analysis was conducted on data exclusively obtained from studies on Asian and European women with PCOS (excluding a solitary South American report from Brazilian), the title should be modified to reflect on this fact (e.g., “A Meta-analysis of Observational Studies on Prolactin Levels in infertile Euro-Asian Women with PCOS”). This meta-analysis report is not about diagnostic features of PCOS! Please rephrase accordingly.

  • Author response and action taken: Thank you for your valuable comment. This study was conducted among women with PCOS, and we did a subgroup analysis for studies with infertile participates (n=6). So all participants of this meta-analysis are not infertile. Furthermore, the included studies in this meta-analysis were not restricted to Euro-Asian, although, the majority of studies was performed in the mentioned region, there are some studies which performed in Africa and America. Therefore we revised the title as follow:” A Meta-analysis of Observational Studies on Prolactin Levels in Women with PCOS”

Abstract:

Line 28: the word “America” should be replaced by “South America” for accuracy and consistency.

  • Author response and action taken: Revised as suggested.

-       Introduction:

o   Lines 54- 58: this is an inaccurate statement on the requirement for excluding other aetiological factors for hyperprolactinemia in women suspected of PCOS. Please rephrase accordingly and provide accurate and adequate citations (current reference #7 (Delcour et al (2019)) cannot be considered in drawing such a major conclusion).

  • Author response and action taken: Thank you for pointing out. We do our best to revise it; hope it meets your expectation.[page 2, lines 53-60]

Lines 66-68: “women” should be replaced with “Euro-Asian women” for consistency and accuracy.

  • Author response and action taken: Thank you for pointing out. We revised it as suggested.

-        Results:

o   Line 130: “America” should be replaced with “South America” for accuracy.

  • Author response and action taken: Edited as suggested.

Lines 138-139: “PCOS patients” should be replaced with “Euro-Asian PCOS patients” for consistency and accuracy.

  • Author response and action taken: Thank you for pointing this out. As above mentioned in cases related to this subgroup we previously added it.

o   Figures 2 and 3:

§  It is hard to read the figures. Please replace them with higher-quality images.

  • Author response and action taken: We added high quality figures.

§  The X-axis needs to be labeled in both figures. It's unclear what is being measured and/or depicted in these forest plots. Please clearly label the X-axis in these plots.

  • Author response and action taken: Thank you for valuable comment. We edit it as suggested.

Lines 147- 154: this is very confusing. It's unclear what exact parameters/covariates are being considered in comparing the Asian and European PCOS patients described in lines 151- 154 and those mentioned in lines 147- 149. Also, it’s unclear what population of African women with PCOS compared their PRL levels with (line 150). Please provide further clarifications and rephrase accordingly.

  • Author response and action taken: lines 147-154 shows the sub-group analysis results. In this sub-group analysis we stratified the studies according the continent and infertility and analysis the PRL levels among PCOS and non-PCOS. We do our best to revise it; hope it meets your expectation

-       Discussion:

o   Lines 170- 174: please remove “across the globe”. Your current meta-analysis does not support the globalization of results.

  • Author response and action taken: In this meta-analysis, although the included studies were performed in different continents, the generalizability of results to the PCOS women around the globe is not possible. Hence, we corrected it.

Lines 172- 173: should read: ‘infertile Euro-Asian patients with PCOS”.

  • Author response and action taken: As mentioned above, all participants are not infertile or from Euro-Asian countries. In statements which are related to the Euro-Asian patients, we pointed out this important.

Lines 175- 192: acknowledging that PCOS is known to alter PRL levels in women, please further discuss the significance of your findings, as well as agreements and disagreements with previous reports accordingly.

  • Author response and action taken: Thank you for your comment. The controversy regarding the PRL levels among PCOS women is presented in the next paragraph [page 12, lines 199-208], through presenting the studies that reported in agreement with our findings, studies with different views, and possible explanations for these controversies.

Lines 202- 203: this is very confusing and contradicts the authors’ conclusions on lines 144- 146 regarding the lack of statistical significance in PRL levels between infertile PCOS and their non-PCOS controls! Please consider rephrasing accordingly.

  • Author response and action taken: It was related to infertile subgroup analysis. However. there is significantly difference among all included studies. This difference was not significant among 6 studies with infertile subjects. We did our best to clarify this in the text.

Lines 230- 236: the authors should further elaborate on the influence of insulin resistance on PRL levels in PCOS women in view of their own meta-analysis (e.g., how insulin resistance might have potentially influenced the outcome of your meta-analysis).

  • Author response and action taken: Thank you for your comment. Changes have been made to further clarify the association between serum PRL concentration and glucose metabolism, and how it might explain the various reports from different regions. [page 13, lines 238-252]

Lines 253- 256: the authors should further discuss and elaborate on plausible factors contributing to the lack of statistical significance in PRL levels among PCOS women reported through the NIH- and AES- based inclusion criteria (e.g., how NIH and AES inclusion criteria influenced outcomes of statistical analyses of PRL levels in infertile PCOS women?).

  • Author response and action taken: Thank you for your feedback. This issue has been discussed in further details based on the differences between different PCOS diagnostic criteria. There is variability between clinical presentations of PCOS according to the different diagnostic criteria. In the studies conducted based on Rotterdam criteria, patients presented with anovulation and polycystic ovaries are also considered to have PCOS, which is not the case in other diagnostic criteria. Moreover, both hyperprolactinemia and PCOS are associated with hyperandrogenism and anovulation, although through separate mechanisms (84). Therefore, the lower diagnosis rate among patients with anovulation in NIH and AES compared to Rotterdam criteria might be the reason why the studies conducted based on these criteria could not reach statistical significance regarding PRL levels. Furthermore, it is proposed that Rotterdam criteria can recognize all possible phenotypic combinations; so it might contribute to differences of findings.  [page 13-14 , lines 275-284]

Reviewer 3 Report

Diagnostic features of Polycystic Ovary Syndrome (PCOS): A Meta-analysis of Observational Studies on Prolactin Levels in Women with PCOS

This paper presents a meta-analysis of observational studies on prolactin levels in women with PCOS, which included 32 observational studies conducted in different geographical regions, in order to prove if there is a difference between prolactin levels in PCOS patients (diagnosed by Rotterdam criteria) and the control group.

The introduction focuses on what we know today about PCOS and hyperprolactinemia and the pathogenic mechanism that lays behind the high levels of prolactin encountered in PCOS women. The information presented is up to date, relevant, and offers the reader a better understanding on the topic.

As far as I am concerned, this paper was conducted on a satisfactory number of studies from different geographical regions. The conclusions drawn by the authors are coherent and significant, supporting the information presented in the listed citations.

The tables and figures presented are easy to interpret and understand.

Good English level.

I recommend it for publication.

Author Response

# Reviewer 3

This paper presents a meta-analysis of observational studies on prolactin levels in women with PCOS, which included 32 observational studies conducted in different geographical regions, in order to prove if there is a difference between prolactin levels in PCOS patients (diagnosed by Rotterdam criteria) and the control group.

The introduction focuses on what we know today about PCOS and hyperprolactinemia and the pathogenic mechanism that lays behind the high levels of prolactin encountered in PCOS women. The information presented is up to date, relevant, and offers the reader a better understanding on the topic.

As far as I am concerned, this paper was conducted on a satisfactory number of studies from different geographical regions. The conclusions drawn by the authors are coherent and significant, supporting the information presented in the listed citations.

The tables and figures presented are easy to interpret and understand.

Good English level.

I recommend it for publication.

Author response: Thank you for your positive feedback and acknowledgment of our manuscripts.

Round 2

Reviewer 2 Report

Thank you for your revisions.